# Molecular and Cellular Characterization of the TH Pathway in the Sea Urchin *Strongylocentrotus purpuratus*

**DOI:** 10.3390/cells12020272

**Published:** 2023-01-10

**Authors:** Maria Cocurullo, Periklis Paganos, Natalie J. Wood, Maria I. Arnone, Paola Oliveri

**Affiliations:** 1Department of Biology and Evolution of Marine Organisms, Stazione Zoologica Anton Dohrn, Villa Comunale, 80121 Naples, Italy; 2Centre for Life’s Origins and Evolution, Research Department of Genetics, Evolution and Environment, University College London, London WC1E 6BT, UK

**Keywords:** Echinoderms, iodine, thyroid hormones, development, skeletal growth

## Abstract

Thyroid Hormones (THs) are a class of signaling molecules produced by coupling iodine with tyrosine residues. In vertebrates, extensive data support their important role in a variety of processes such as metabolism, development and metamorphosis. On the other hand, in invertebrates, the synthesis and role of the THs have been, so far, poorly investigated, thus limiting our understanding of the function and evolution of this important animal signaling pathway. In sea urchins, for example, while several studies focused on the availability and function of external sources of iodotyrosines, preliminary evidence suggests that an endogenous TH pathway might be in place. Here, integrating available literature with an in silico analysis, various homologous genes of the vertebrate TH molecular toolkit have been identified in the sea urchin *Strongylocentrotus purpuratus*. They include genes involved in the synthesis (*Sp-Pxdn*), metabolism (*Sp-Dios*), transport (*Sp-Ttrl*, *Sp-Mct7/8/10*) and response (*Sp-Thr*, *Sp-Rxr* and *Sp-Integrin αP*) to thyroid hormones. To understand the cell type(s) involved in TH synthesis and/or response, we studied the spatial expression of the TH toolkit during urchin development. Exploiting single-cell transcriptomics data in conjunction with in situ hybridization and immunohistochemistry, we identified cell types that are potentially producing or responding to THs in the sea urchin. Finally, growing sea urchin embryos until the larva stage with and without a source of inorganic iodine, we provided evidence that iodine organification is important for larval skeleton growth.

## 1. Introduction

Thyroid hormones (THs) are a group of signaling molecules produced by the coupling of iodine with tyrosine residues; however, the term THs is often used to indicate two specific compounds: triiodothyronine (T3) and thyroxine (T4). Thyroid hormones are multifunctional, and an evolutionarily conserved role in metabolism, development and metamorphosis has been reported (for reviews, see [1,2,3]). In vertebrates, THs are produced by a distinct organ, the thyroid gland, in a complex process well described in [4]. Briefly, iodine is accumulated inside the thyroid gland by a sodium/iodide symporter (NIS). Once in the lumen of the gland, the thyroid peroxidase (TPO) catalyzes the oxidative coupling of iodine to tyrosine residues of the thyroglobulin (Tg), and this process requires H_2_O_2_. Then, upon stimulation, the iodinated tyrosine residues are cleaved to free T4 and T3, which then enter the bloodstream. Finally, THs are transported by albumin, thyroxine-binding globulin (TBG) and transitherin (TTR) to the target tissues, which they can enter in various ways. Being lipophilic molecules, it was initially hypothesized that THs could cross the plasma membrane by passive diffusion; however, it has been shown that this process can also be facilitated by specific transporters such as the monocarboxylate transporter 8 (MCT8) [5].

The TH signaling cascade is mediated by the Thyroid Hormone Receptors (TRs). TRs belong to the class of nuclear receptors. Being able to directly bind the DNA, this class of receptors is classified as transcription factors. In general, nuclear receptors bind the DNA only when the ligand is present, activating or inhibiting the transcription of the target genes. Interestingly, TRs bind the DNA even when the ligand is not present. In this case, the TRs recruit transcriptional repression complexes inhibiting the transcription. When the receptor binds the hormone, the complex is disassembled, and the receptor forms a new heterodimer with the Retinoic X Receptor (RXR) that binds the target genes activating their expression [6,7]. Notably, while T3, T4 and other members of the THs all have distinct functions [8,9], T3_,_ which has a higher affinity for the receptor, is often considered to be the sole active form of the THs [10]. In addition to the canonical pathway mentioned above, non-canonical modes of action have also been described (reviewed in [11]). For instance, it has been reported that T3 and T4 can both activate the transduction cascade mediated by the integrin αvβ3, which in turn activates MAPK, PI3K and nitric oxide synthase. Interestingly, this has been described to be operating during T4-induced angiogenesis [12,13].

TH production in the thyroid gland is regulated by the Thyroid-Stimulating Hormone (TSH) produced by the pituitary gland. TSH, in turn, is produced upon stimulus from the hypothalamus through the Thyrotropin-Releasing Hormone (TRH). Moreover, a group of enzymes called Deiodinases (DIOs) can convert T4 into T3 and other THs compounds by removing iodine residues. So far, three types of DIOs, DIO1, DIO2, and DIO3, have been described to be essential for these processes, while the use of the specific enzyme depends on which residue is involved during the deiodinase reaction. The result of the deiodinase activity is to finely regulate the bioactivity of THs in a tissue-specific manner. An example of the importance of this regulative process can be found during amphibian metamorphosis, where DIOs expression levels and activity were shown to be finely regulated in a tissue and stage-specific manner [14]. 

Outside vertebrates, the synthesis and role of the THs are less clear and poorly investigated. Despite the fact that the thyroid gland as a distinct organ is a novelty of vertebrates, it has been shown that other chordates [6,15], some invertebrates [16], as well as non-animal representatives such as algae [17,18,19], which lack a proper thyroid organ, are also able to produce and/or to respond to THs (for reviews see [20,21,22]). Nonetheless, thyroglobulin was suggested to be a vertebrate novelty; therefore, the mechanism by which THs are produced outside this taxon is still a mystery [23]. Moreover, the endostyle, an organ dedicated to TH production, is found both in non-vertebrate chordates, such as amphioxus, and in the vertebrate chordate sea lamprey at the larval stage [24,25,26]. Interestingly, both thyroid and endostyle are closely linked with the pharynx region, suggesting a possible link between the two. For these reasons, the endostyle is commonly considered to be a thyroid homolog. Intriguingly, it seems that the endogenous organification of iodine into iodotyroiodines is not a unique source of iodinated compounds for animals. In fact, most invertebrates cannot produce THs by themselves and rely on external sources of iodinated compounds such as algae (by ingestion) [10,27]. 

Echinoderms represent an interesting case study. Although there is evidence suggesting the presence of an internal pathway for THs synthesis [16,27], it is most commonly believed that their source of iodotyrosines comes from diet, and these are used as an indicator of nourishment [10,28]. Furthermore, it has been suggested that the acquisition of an endogenous pathway for THs production was important for the emergence of lecithotrophy in Echinoderms [29]. In the case of sea urchin larvae, Heyland and collaborators in 2006 [16] suggested that the sea urchin *Lytechinus variegatus* is able to uptake iodine, possibly through LvTPO. A peroxidase-dependent mechanism for the iodine uptake was subsequently found also in the species *Strongylocentrotus purpuratus* [30]. To assess the role of THs during sea urchin development, the same authors exposed pluteus larvae to various concentrations of T4 and measured the spicules’ length. Their data suggest that exposure to T4 results in decreased arm growth at the early larval stage, while it induces the metamorphosis of competent larvae. However, in 2018 Taylor and Heyland, applying a similar approach involving measurement of skeletogenesis initiation rates at the gastrula stage, showed that THs accelerate the initiation of the skeletogenic process in *S. purpuratus* larvae [31]. Moreover, their data suggest that T4 activity is possibly mediated by a non-canonical transduction pathway involving integrins and MAPK (ERK 1/2), a mechanism similar to the one employed during T4-induced angiogenesis in humans. Lastly, Sainath and colleagues in 2019 [4] identified several gene homologues involved in the TH pathway to be encoded in the genome of the sea urchin *S. purpuratus*. However, the functionality of these genes remains elusive. Taken together all these data, it is evident that TH signaling is ancient and has a crucial role in sea urchin embryogenesis and larval growth, while the mechanism by which THs facilitate this remains a mystery. Can sea urchin larvae produce THs? How are THs regulating larval growth and what is their possible function? Which are the key players and where are they expressed? What’s the role of iodine?

In this study, we investigate the genes of the TH signaling cascade encoded in the sea urchin genome and where they are expressed during late embryogenesis and early larval development. To this end, we performed in silico analysis of genes involved in the TH signaling pathway and found several vertebrate homologs to be encoded in the *S. purpuratus* genome. Using single-cell transcriptomics paired with in situ hybridization and immunohistochemistry, we are able to identify cell-type families that, based on their transcriptomic signature, are potentially producing or responding to THs in sea urchins. Lastly, we investigate how the presence of iodine in the embryo micro-environment affects larval development and growth by rearing *S. purpuratus* larvae in Artificial Sea Water (ASW) made without or with a supplement of an iodine source (Sodium iodide, NaI). Our results show that larvae grown without iodine supplementation have significantly shorter skeletal rods, suggesting that endogenous organification of iodine is taking place and highlighting the importance of organification in proper skeletal growth.

## 2. Materials and Methods

### 2.1. Animal Husbandry

Adult *S. purpuratus* were obtained from Patrick Leahy (Kerckhoff Marine Laboratory, California Institute of Technology, Pasadena, CA, USA) and maintained in circulating seawater at Stazione Zoologica Anton Dohrn in Naples (SZN) and at University College of London (UCL) at 14 °C.

### 2.2. Culturing of Embryos and Larvae

Gametes were obtained by vigorously shaking the adult sea urchins. The sperm was collected dry using a Pasteur pipette and stored at 4 °C until usage. To collect eggs, females were inverted over a beaker filled with FASW (at UCL) or FMSW (at SZN). About 20 mL of eggs were fertilized, adding a few drops of sperm diluted 1:10,000. Embryos were transferred in FMSW or FASW and reared at 15 °C under a 12 h light/12 h dark or dark-only cycle. For iodine experiments, embryos were reared as described in Section 2.6.

### 2.3. In Silico Identification of Genes of Interest 

A BLAST search of known human protein components of the TH pathway was performed against *S. purpuratus* using both NCBI (https://blast.ncbi.nlm.nih.gov/Blast.cgi, accessed between June–September 2021) and Echinobase. Echinobase (available at https://new.echinobase.org/entry/, accessed between June–September 2021) is a web-based resource that gives access to genomic, expression and functional data of various Echinoderms [32]. The E-value threshold of 10^−10^ was used as a significant value. 

### 2.4. Whole Mount RNA Fluorescent In Situ Hybridization (FISH) and Immunohistochemistry (IHC)

Whole-mount RNA fluorescent in situ hybridization and combined FISH-IHC were performed as described in [33,34]. Briefly, specimens at different stages of development were collected and fixed in 4% PFA in 0.1 M MOPS and 0.5 M NaCl for at least one night at 4°C. Then, they were washed several times with MOPS buffer (0.1 M MOPS, 0.5 M NaCl, 0.01% Tween 20) for 15 min at RT, dehydrated in 70% ethanol and finally stored at −20 °C until usage. Antisense probes were transcribed from linearized DNA and labeled either during transcription using Digoxigenin (Roche) or Fluorescein (Roche) labeled ribonucleotides following the manufacturer’s instructions. Primer sequences used for cDNA isolation and probes synthesis are in Appendix A. Specimens were left to hybridize in the probe containing hybridization buffer (0.3–0.5 ng/μL) overnight at 65°C. A fluorescent signal was developed using fluorophore-conjugated tyramide technology (Akoya Biosciences, Cat. #NEL752001KT). For combined FISH-IHC, after the tyramide amplification step, samples were incubated in blocking (containing 1 mg/mL Bovine Serum Albumin and 4% sheep serum in PBS) for 1 h at RT, then transferred in Anti-Msp130 (gift from Dr. Chuck Ettensohn (6a9, [35]), diluted 1:50) for 1 h and 30 min at 37 °C. Anti-Msp130 was used to label skeletogenic cells. Samples were washed 4–6 times with PBS 1×, then stained with Alexa Fluor secondary antibodies (488 mice) diluted 1:1000 in blocking, and finally washed 4–6 times with PBS 1x. DAPI was added to the samples at a final dilution of 1:10000 to stain nuclei. Specimens were imaged using a Zeiss LSM 700 confocal microscope, and pictures were analyzed using ImageJ.

### 2.5. Identification of the Expression Profiles of the TH Pathway Components through Scrna-Seq

The expression pattern of the sea urchin orthologues involved in the TH pathway that were identified by our in silico analysis was reconstructed using available scRNA-seq data. The profiles of TH genes were investigated at 2 developmental time points: 2 dpf gastrula (unpublished data, Arnone) and 3 dpf early pluteus larva [36] and were represented as dotplots. The dotplots were generated in R (version 4.1.1) using the DotPlot function incorporated into the Seurat R package (version 4.0.2) [37]. Each dotplot shows the average expression of the genes of interest in distinct cell type families as well as the percentage of cells expressing them.

### 2.6. Iodine Experiments

*S. purpuratus* embryos were cultured in 0.2 μm filtered artificial seawater with a pH of 8.2 and a salinity of 34.5 ppt (FASW 1L: 28.3 g NaCl; 0.77 g KCl; 5.41 g MgCl_2·_6H_2_O; 3.42 g MgSO_4_; 0.2 g NaHCO_3_; 1.56 g CaCl_2_) or sodium iodide (NaI; 57 μg/L) FASW in the presence of antibiotics from fertilization (0 hpf) until pluteus larval stage (4 days post fertilization). Embryos were transferred to freshly made FASW at least once every 2 days. At 4 dpf larvae, the length of larval skeletons was measured. Larvae were mounted under a 22 by 22 mm coverslip, so their skeletons were completely flattened but not broken. Brightfield images were taken of larval skeletons, and the length was measured in pixels using the length measurement option in ImageJ software (NIH). The 4 distinct spicules of the larval skeleton were separately measured, the body rod, post-oral, oral transverse, and oral distal spicules. Ggplot in RStudio was used to put the skeleton length data on violin plots with box and whisker plots overlaid. The software also performed Student’s 2-tailed *t*-test to determine the statistically significant difference in larval skeleton length between the 2 conditions.

## 3. Results

### 3.1. Identification of the TH Pathway Toolkit Encoded in the Sea Urchin Genome

A literature survey in combination with a genome search using human protein sequences was conducted to identify the molecular components present in the *S. purpuratus* genome that may be involved in the synthesis, metabolism and response to THs. Table 1 summarizes the results of the genomic and literature search.

Concerning genes implicated in vertebrate TH synthesis, our search and analysis identified putative sea urchin orthologs of the Iodine transporter NIS (LOC576134, *Sp-Slc5a5*) and the TPO enzyme (LOC593243, *Sp-Pxdn*) involved in the TH biosynthetic pathway. Interestingly, while our genome survey did not produce any results on the putative sea urchin homolog of Tg, phylogenetic analysis by Belkadi and colleagues [39] identified a putative Tg homolog in *S. purpuratus* (LOC755632), which encodes for Thyroglobulin type-1 repeats (TY). Moreover, in the newest *S. purpuratus* genome 5.0, another gene (LOC589650) has been annotated as *Sp-Tg*. However, there is no evidence of the expression of this gene from transcriptomic data (Appendix A). 

Regarding genes that have been known to participate in the signaling cascade initiated by TH hormones in vertebrates, we identified sea urchin orthologs of Thyroid hormone receptor TR (LOC584535, *Sp-Thr*), the TH transporters TTR (LOC589020, *Sp-Ttrl*) and MCT8 (LOC576836, *Sp-Mct7/8/10*), the TH activity mediator DIO (LOC577015, *Sp-Dio*) and integrin αvβ3 (LOC373206, *Sp-Integrin αP*). The *Sp-Integrin αP* gene corresponds to the one suggested by Taylor and Heyland [31] as a mediator of the TH regulation of skeletal growth. Moreover, Susan and collaborators in 2000 [42] found this integrin to be expressed in *S. purpuratus* at low levels at early embryonic stages and at higher levels from the gastrula stage, with the highest expression at the pluteus larva stage. 

Looking more in detail at the canonical pathway, it appears that while vertebrate genomes encode for two TRs genes (TRα and TRβ) emerging from genome duplication at the basis of vertebrates [28], *S. purpuratus* has only one TR gene (*Sp-Thr*) that was previously annotated as two separate gene models. The phylogenetic analysis performed by Howard-Ashby and colleagues in 2006 [41] grouped the *Sp-Thr* together with the *Hs-Thra* and *Hs-Thrb*, confirming the homology. 

Of the three proteins acting as carriers of THs in the bloodstream, we could find only an *Sp-Ttrl*, sharing sequence homology with the human transthyretin precursor. For the other two carriers (ALB and TBG), no homologs have been reported in the literature. Nonetheless, a sea urchin protein known as Endo16, whose function is unknown, has been suggested to have originated from an ancestral albumin gene through a duplication event [4,40].

Taken together, the genome survey identified genes related to all components of the TH pathway, including genes that might have a role in the THs synthesis (*Sp-Tg*, *Sp-Pxdn*), metabolism (*Sp-DIOs*), transport (*Sp- Mct7/8/10*, *Sp-Ttrl*) and response (*Sp-Thr*, *Sp- Integrin αP*) are present in the sea urchin *S. purpuratus* genome. 

### 3.2. Spatiotemporal Expression Pattern of Putative Sea Urchin TH Pathway Components

Once we identified the sea urchin homologs of the vertebrate TH pathway components, the expression of the sea urchin genes was investigated using publicly available transcriptomic data for temporal expression and single-cell transcriptomics [36] paired with FISH and IHC to understand the spatial expression of the TH toolkit during sea urchin development and in the larvae. 

The temporal expression is reported in the Appendix A and shows most of the genes dynamically expressed during the development of the sea urchin embryo and in the larvae. For the spatial expression, two developmental time points were analyzed: 2 dpf (days post fertilization; Figure 1 and Appendix A) gastrula stage (which represents a late embryonic stage in which cell specification has been completed and cell type differentiation takes place, e.g., skeletal differentiation) and the free swimming and feeding 3 dpf pluteus larva stage (Figure 1 and Appendix A). Plotting for the genes identified in the *S. purpuratus* scRNA-seq datasets from Paganos and collaborators [30] revealed that most of these genes are expressed in both developmental time points *(peroxidasin, transthyretin, monocarboxylate transporter 10, thyroid hormone receptor, integrins alpha-V, retinoic X receptor,* and *deiodinase type I)* suggesting the presence of primary machinery that is already in place and operating in specific cell type families during gastrula and pluteus stages (Appendix A).

Next, we focused on the expression domains of three key components of the TH pathway, *Sp-Pxdn, Sp-Dio, and Sp-Thr*, found to be in distinct cell type families in both developmental time points. Plotting the average expression of *Sp-Pxdn* at the gastrula stage (Figure 1A,C) showed that *Pxdn* transcripts are predicted to be expressed in several cell types, including mesodermally derived immune cell populations [38] (globular, pigment and blastocoelar cells), while its predominant expression is predicted to be in skeletogenic cells (PMCs). Moreover, significant expression levels in endodermally-derived cell type families, such as the midgut and foregut domains, are predicted. Similarly, at the pluteus stage (Figure 1(E,F)), *Sp-Pxdn* transcripts are predicted to be expressed in the same mesodermally derived cell type families, with a predominant expression in PMCs also having three additional domains of expression including the coelomic pouches, parts of the oral ectoderm surrounding the mouth of the larva, and the anal sphincter.

FISH using a specific *Sp-Pxdn* antisense probe designed against the transcript that showed the highest expression (WHL22.6892) was carried out at both developmental time points and showed expression of *Pxdn* in cell types consistent with the scRNA-seq predictions. In detail, *Sp-Pxdn* is found to be strongly expressed in the foregut region of the 2 dpf gastrula embryo (Figure 1(B,B1) and the esophageal region of the 3 dpf pluteus larva (Figure 1(D,D1)). Among the *Sp-Pxdn* positive mesodermally derived cell type families spread throughout the blastocoel of the embryo and larva, significant expression is detected in skeletal cells as shown by the co-localized signal of *Sp-Pxdn* and the skeletogenic marker Msp130 [35]. Interestingly, at the gastrula stage, *Sp-Pxdn* seems to be expressed in a larger skeletal cell population, while at the pluteus stage, its expression appears to be limited to the cells at the vertex of the larva.

Out of the several *deiodinase-*encoding genes that are present in our analysis (Table 1), only one of them (SPU_002251) seems to be constitutively expressed at both developmental stages. The average expression of *Sp-Dio* (Figure 1G,I,L,K) showed that, at both developmental time points, its expression is enriched in the aboral ectoderm region. At the gastrula stage, *Sp-Dio* is also predicted to be expressed in the developing ciliary band, oral ectoderm, and exocrine midgut regions, while, at the pluteus stage, low levels of *Sp-Dio* are predicted to be present in a broad spectrum of cell types families including ciliary band, apical plate, neurons, immune cells, intestine, anal sphincter and the stomach of the larva. The scRNA-seq predictions are confirmed by FISH experiments (Figure 1(H,H1,J,J1)) that show a predominant expression of *Sp-Dio* in the aboral ectoderm of the embryo and larva while revealing an overall increase in the *Sp-Dio* positive domains at pluteus stage.

Moreover, we characterized the only homolog of the thyroid hormone receptor present in our analysis and to do so, we generated an antisense probe recognizing the transcript with the highest level of expression (WHL22.211956). Similar to *Sp-Pxdn*, *Sp-Thr* is predicted to be expressed in a broad spectrum of mesodermally derived cell type families. At the gastrula stage (Figure 1M,O), *Sp-Thr* transcripts are predicted to be expressed in muscle and coelomic pouch progenitors in all immune-related cell type families (blastocoelar, pigment and globular cells) as well as in skeletal cells. Additionally, the ectodermal domains of the anterior neuroectoderm and oral ectoderm appear to be *Sp-Thr* positive. Interestingly, at the pluteus stage (Figure 1R,Q), while *Sp-Thr* expression remains in most domains such as muscles, coelomic pouches, and immune and blastocoelar cells, it appears to be silenced in skeletal cells and apical plate domains, suggesting an early role of *Sp-Thr* in the differentiation of these cell types. At the same time, scRNA-seq predicts that *Sp-Thr* expression is upregulated in two novel domains: the cardiac and anal sphincters. FISH using an *Sp-Thr* antisense probe (Figure 1(N,N1,P,P1)) revealed similar expression patterns to the ones predicted by scRNA-seq. At the gastrula stage (Figure 1(N,N1)), *Sp-Thr* transcripts were detected in various cell populations within the blastocoel of the embryo, some of which appear to be PMCs as shown by the co-localized signal of the *Sp-Thr* probe and the immunohistochemical detection of Msp130. At the pluteus stage (Figure 1(P,P1)), *Sp-Thr* transcripts seem enriched in the coelomic pouches of the larva, a population of cells that, based on their localization, appear to be consistent with a mesodermal origin, possibly pigment or blastocelar cells, as well as in distinct cells of the apical organ, whose distribution suggests that they could be the serotonergic neurons of the larva.

Once it was established that the scRNA-seq data could faithfully predict the spatial expression data, the genes of the TH toolkit identified in the in silico approach were analyzed for co-expression. We focused on the *Sp-Rxr,* assuming that its colocalization with *Sp-Thr* would indicate in which cell types the canonical TH signaling could be in place. Noticeably, the average expression from scRNA-seq data predicts the coexpression of THR-RXR in skeletogenic cells at 2 dpf (Appendix A) but not at 3 dpf (Appendix A). Also, *Sp-Thr* and *Sp-Rxr* are predicted to coexpress in the coelomic pouch cell type family at 3 dpf.

To further investigate the hypothesis, suggested by [31], for a major role of the non-canonical mechanism mediated by integrins in THs control of skeletogenesis, we interrogated single-cell data for the *Sp-IntegrinaP* which shares sequence homology with the human integrin αv subunit). Intriguingly, at 2 dpf (Appendix A) but not at 3 dpf (Appendix A), the *Sp-Integrin αP* is predicted in the skeletogenic cells. Finally, we focused our attention on the MCT10 homolog *Sp-Mct7/8/10*. At gastrula, the average expression plot predicts *Sp-Mct7/8/107* to be expressed in globular, blastocoelar and midgut cells (Appendix A). Then, at the pluteus stage, it expands into many more domains, including the apical plate, stomach, intestine, and anus (Appendix A). 

Taken together, our expression data suggest the transcription of many of the genes involved in the synthesis of and response to THs during late embryonic and early larval development. The expression of these genes appears to be dynamic during development and cell type family specific. This suggests diverse functions of the TH pathway(s) across cell types at a critical developmental window when cell type differentiation occurs. 

### 3.3. Role of Iodine during Development

The gene analysis and gene expression data suggest that the organification of iodine might have a role in sea urchin development. To investigate the function of iodine depletion during the early developmental stage, we reared sea urchins from fertilization (0 hpf) to pluteus larval stage (4 dpf) in FASW (lacking an additional iodine source), and FASW supplemented with NaI. The amount of NaI to add was calculated based on the average iodine concentration estimated in the ocean [43]. At 4 dpf, we measured the length of each part of the larval skeleton, and data showed that larvae reared in NaI FASW had longer skeletons than larvae grown in FASW only (Figure 2A). In particular, *S. purpuratus* larvae reared in NaI FASW had significantly longer post-oral, oral distal, and oral transverse spicules (Figure 2A,B), while body rod spicules were slightly, but still significantly longer. A schematic representation of these findings is depicted in Figure 2C. Overall, the finding of an NaI-mediated phenotypic response of the larvae, in association with specific expression of the TH pathway genes in skeletal cells during development, highlights an important role of iodine, and likely THs-mediated signaling, in skeletal development and regulation of the larval growth. 

## 4. Discussion

### 4.1. Key Components of the TH Pathway Are Present in the Sea Urchin Genome

Despite plenty of evidence suggesting that THs play a role in sea urchin skeletal development and metamorphosis, little is known about the molecular components encoded in the sea urchin genome and their functionality. To fill this gap, we first performed a genome survey and took advantage of available resources to identify key components of the TH pathway encoded in the sea urchin *S. purpuratus* genome (for a summary of our findings, see Figure 3).

First, we focused on the molecular machinery involved in the synthesis of THs. The initial step for iodine organification is its transport and accumulation inside the producing tissue/cell. In vertebrates, iodine is accumulated in the thyroid by the NIS. Nonetheless, previous evidence based on pharmacological treatments suggested that the NIS transporter system is not involved in the iodine uptake in sea urchins [16,30]. Future perturbation experiments of the sodium/iodine symporter (LOC576134) identified in the sea urchin genome are needed to understand its function in sea urchins as well as its evolutionary origins. The second step of iodine organification involves the protein Thyroglobulin. The existence of a Tg in sea urchins is still elusive. Nonetheless, a possible homolog was reported by [39] (LOC755632), and another gene is annotated as *Sp-Tg* in Echinobase (LOC589650). However, none of these two genes were found expressed in our scRNA-seq, which corroborates with their very low expression levels as indicated by bulk-embryo developmental transcriptomes (Appendix A). This sets the bases for the possible identification and characterization of an *S. purpuratus* thyroglobulin homolog. Lastly, we identified the *Sp-Pxdn* (LOC593243) as a possible homolog of the human TPO, an enzyme responsible for the THs synthesis, providing evidence that a mechanism for iodine organification in *S. purpuratus* might exist. Moreover, the existence of many *Sp-DIOs* (Table 1) suggests that a mechanism for THs metabolism and tissue-specific regulation of their bioactivity might be in place also in sea urchins. In our experiment, we were able to characterize the expression pattern of one of these DIOs, and this is further discussed later. Intriguingly, the *S. purpuratus* genome encodes for at least two genes, *Sp-Ttrl* and *Sp-Mct7/8/107*., which might be involved in THs transport. Finally, the existence of *Sp-Thr* and *Sp-Rxr,* as well as the *Sp-Integrin αP*, suggests that both canonical and non-canonical transduction cascades in response to THs might occur in sea urchins.

### 4.2. Possible Mechanism of TH Production and Function

The expression of the enzymes involved in TH biosynthesis and metabolism (*Sp-Pxdn, Sp-DIOs*) suggests that a mechanism for endogenous organification of iodine is present in sea urchin embryos and larvae. The expression data suggest that many cell types may be responsible for TH production, in particular, skeletogenic (at both stages) and coelomic pouch cells at 3 dpf. This is intriguing, given the fact that the left coelomic pouch will eventually give rise to the adult rudiment).

Importantly, larvae reared in FASW (which is poor in iodine) had significantly shorter skeletons compared to plutei grown in FASW containing NaI (Figure 2). This suggests that endogenous iodine organification is important for skeletal growth in the sea urchin *S. purpuratus*, in contrast with the predominant idea that the main TH pathway relies on the exogenous supply of organic iodine, i.e., obtained from food [10,22]. 

Moving on to the expression pattern of *Sp-Dio*, this remains hard to interpret; its main expression in the aboral ectoderm and in the apical plate region at 3 dpf suggests some interesting scenarios. First, the enrichment of *Sp-Dio* in the aboral ectoderm is adjacent to the *Pxdn*-Msp130 positive and the *Thr* positive skeletal cells in the vertex (Figure 1E,K), suggesting a paracrine effect. Furthermore, the expression pattern from in situ indicates that only one coelomic pouch expresses deiodinase, suggesting a differential metabolism (and therefore a different effect) of the THs on the left and right coelomic pouches. This observation is intriguing in light of the fact that only the left coelomic pouch will give rise to the adult rudiment and that THs are known to accelerate development and metamorphosis [16,31]. Another interesting domain of expression concerns the localization of *Thr* and *Dio* probes in the apical plate region (Figure 1E,K). In this territory, the larval apical organ is situated, which consists of a group of sensory serotonergic cells [44]. Functional data indicate a role for serotonin in the modulation of ciliary beating, pyloric sphincter opening and metamorphosis [44,45,46,47]. Although at this stage we have expression data for only one DIO, nonetheless, it is possible that the other DIOs identified in the sea urchin genome are functional at later stages of development. Importantly, the plasticity of the *Sp-Dio* expression at gastrula and pluteus stages and the existence of other DIOs in the sea urchin genome suggests that TH metabolism is finely regulated to exert differential function on different tissues and stages, a scenario similar to the events taking place during amphibian development and metamorphosis. 

Moreover, even though we did not characterize the expression pattern of *Sp-Mct7/8/10* through FISH, scRNA-seq data suggest an interesting association of this gene with the gut and cells involved in the immune response (globular, blastocoelar cells at 2 dpf). At the pluteus stage, the expression of this gene is predicted in many domains associated with the gut. This last piece of evidence suggests a possible role for *Sp-Mct7/8/10* in the uptake of THs from food (at later stages) and might be interesting to investigate further.

### 4.3. TH Pathway Regulating Skeletal Growth and Metamorphosis 

The sea urchin gastrula stage represents an important step for larval development, as skeletogenic cells are arranged in their positions and start to deposit calcium carbonate (CaCO_3_) at this stage [48]. Taylor and Heyland [31] showed that skeletogenic initiation is accelerated by the supply of THs through integrin-mediated signaling. Interestingly, our data suggest that skeletogenic cells at 2 dpf might respond to the THs both through the canonical signaling pathway (scRNA-seq data predict co-expression of *Sp-THR* and *Sp-RXR*) and a non-canonical, integrin-mediated pathway (mechanisms suggested by [31]). In contrast, at a later stage, both *Sp-Thr and Sp-Integrin αP* are not any more expressed in skeletogenic cells. Overall, our data provide more evidence for the role that THs play in regulating skeletogenesis, previously published by Taylor and Heyland [31], and suggest a differential regulation at specific developmental stages. It is worth mentioning that the sea urchin larval skeleton features a well-characterized case of phenotypic plasticity in response to food availability. Interestingly, the various components (rods) of the larval skeleton display different levels of plasticity that correlate with their anatomical function. In particular, the post-oral (PO) and oral distal + oral transversal (OD + OT) spicule rods, which are responsible for the support of the ciliated epithelium covering the larval arms, display the strongest response. Indeed, sea urchin larvae utilize ciliary beating both to swim and to collect food, and the number of cilia is proportional to arm length. When the food is scarce, larvae develop longer PO and OD + OT spicule rods in order to increase the surface of ciliated ectoderm and therefore capture food more efficiently. On the contrary, body rods (BR) are not affected [49,50]. Our larval growth experiments demonstrate that, without iodine supplementation, larvae cannot grow as long skeletal rods as they would have if provided with a proper iodine source. Noteworthy, this effect is visible not only on PO and OD + OT spicule rods but also on the BR, suggesting that it corresponds to a general impairment of the skeletal growth rather than a secondary physiological response. Taken together, our results show that iodine and TH signaling plays an important role in larval growth and highlight the fact that the skeleton is a key target of environmental control. How the different signaling pathways, developmental mechanisms and environmental cues crosstalk is still unknown and needs to be further investigated.

Another interesting finding is the predicted co-localization of *Sp-Thr* and *Sp-Rxr* in the coelomic pouches and the expression of *Sp-Dio* only in one of them. In fact, in sea urchins, the left coelomic pouch is the one that will eventually give rise to the adult rudiment [51], another process shown to be influenced by THs. Notably, the scRNA-seq data show that *Sp-Thr* is expressed in more cell type families than *Sp-Rxr* (such as globular and pigment cells at 2 dpf, immune and esophageal muscles cells at 3 dpf), suggesting that other canonical and non-canonical signaling pathways described in [11] might be in place in these territories. Finally, an interesting group of cell types emerged from our analysis: in particular, blastocoelar, globular, and immune cells, which are linked with the larval immune system [38]. The presence of many components of the TH pathway in these cells suggests that THs might have an important function in their response. 

## 5. Conclusions

Overall, our data suggest that the key components of the TH pathway are encoded in the sea urchin *S. purpuratus* genome. Gene expression analyses unravel a complex pattern of expression for each putative component of the sea urchin machinery at late embryonic developmental and pluteus larval stages. In particular, our analysis provided insights into how the TH machinery might function to regulate the known roles in skeletogenesis and metamorphosis and expand our understanding of the mechanism by which THs might induce distinct phenotypic responses at different stages of development. Moreover, we found a previously unknown association of various components of the sea urchin TH machinery with cell types such as blastocoelar and immune cells. This suggests previously unknown roles of the THs in sea urchins, which might be investigated in further studies. 

Importantly, growing larvae in ASW with and without a NaI showed that iodine increases skeletal growth in the sea urchin larva, suggesting that a mechanism for iodine organification is in place in this organism and that the mechanism affects skeletal growth at the stages investigated. This is consistent with the fact that the endogenous production of iodotyrosines seems to be a more likely source of these signaling molecules at early developmental stages when the embryos do not yet possess the structural and molecular machinery to introduce and digest food, and, therefore, an exogenous source of THs at early gastrula stage looks hard to explain. Nonetheless, at later stages, food might become an important source of iodotyrosines to guide the development of the sea urchin larvae through adult rudiment formation and metamorphosis, which is known to require adequate signaling from the environment [52]. 

## Figures and Tables

**Figure 1 cells-12-00272-f001:**
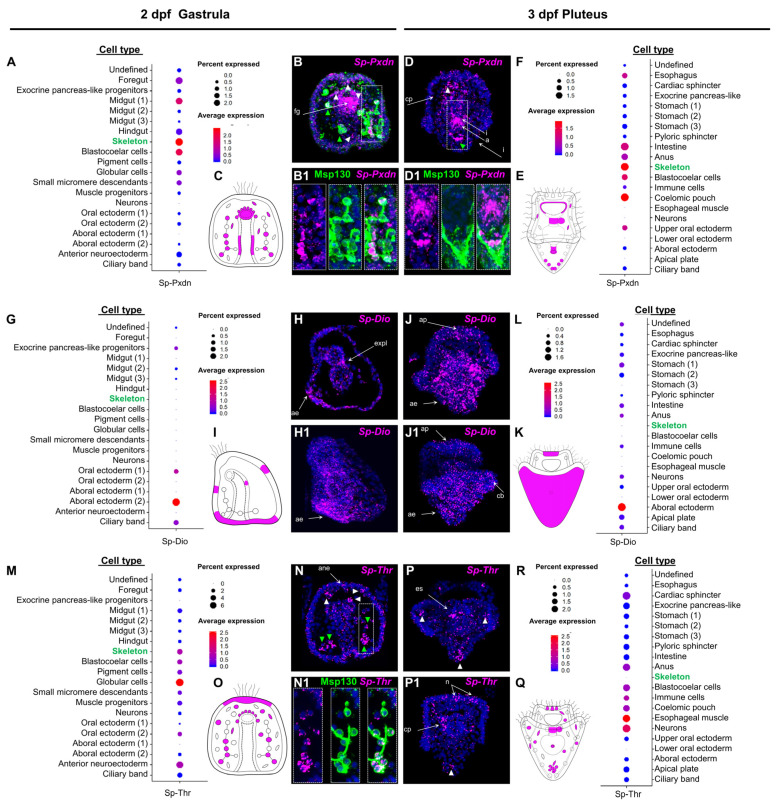
Spatiotemporal expression pattern of putative sea urchin TH pathway components. (**A**,**F**,**G**,**L**,**M**,**R**) Dotplot showing the average expression and percentage of cells per cluster containing transcripts for *Sp-Pxdn* at 2 dpf, *Sp-Pxdn at 3 dpf, Sp-DIO* at 2 hpf, *Sp-DIO* at 3 dpf, and *Sp-Thr* at 2 dpf, *Sp-Thr* at 3 dpf, respectively. (**B**,**B1**,**D**,**D1**) Coupled FISH-IHC showing the expression of *Sp-Pxdn* in PMC, as indicated by the colocalization of *Sp-Pxdn* transcripts and the immunoreactivity-based signal of the skeletal marker Msp130 at 2 and 3 dpf. White triangles= blastocoelar cells. Green triangles= skeletogenic cells. (**H**,**H1**,**J**,**J1**) FISH showing the expression pattern of *Sp-DIO* at 2 and 3 dpf. (**N**,**N1**,**P**,**P1**) coupled FISH-IHC showing the expression patterns of *Sp-Thr* and Msp130 at 2 and 3 dpf. White triangle= blastocoelar cells. (**C**,**E**,**I**,**K**,**O**,**Q**) Schematic representations of the expression domains of *Sp-Pxdn* at 2 (**C**) and 3 (**E**) dpf, *Sp-Dio* at 2 (**I**) and 3 (**K**) dpf, and *Sp-Thr* at 2 (**O**) and 3 (**Q**) dpf as illustrated by the dotplots. a= anus; ae= aboral ectoderm; ane = anterior neuroectoderm; ap= apical plate; cp = coelomic pouch; es = esophagus; expl = exocrine pancreas-like; fg = forefut; I = intestine; n = neurons.

**Figure 2 cells-12-00272-f002:**
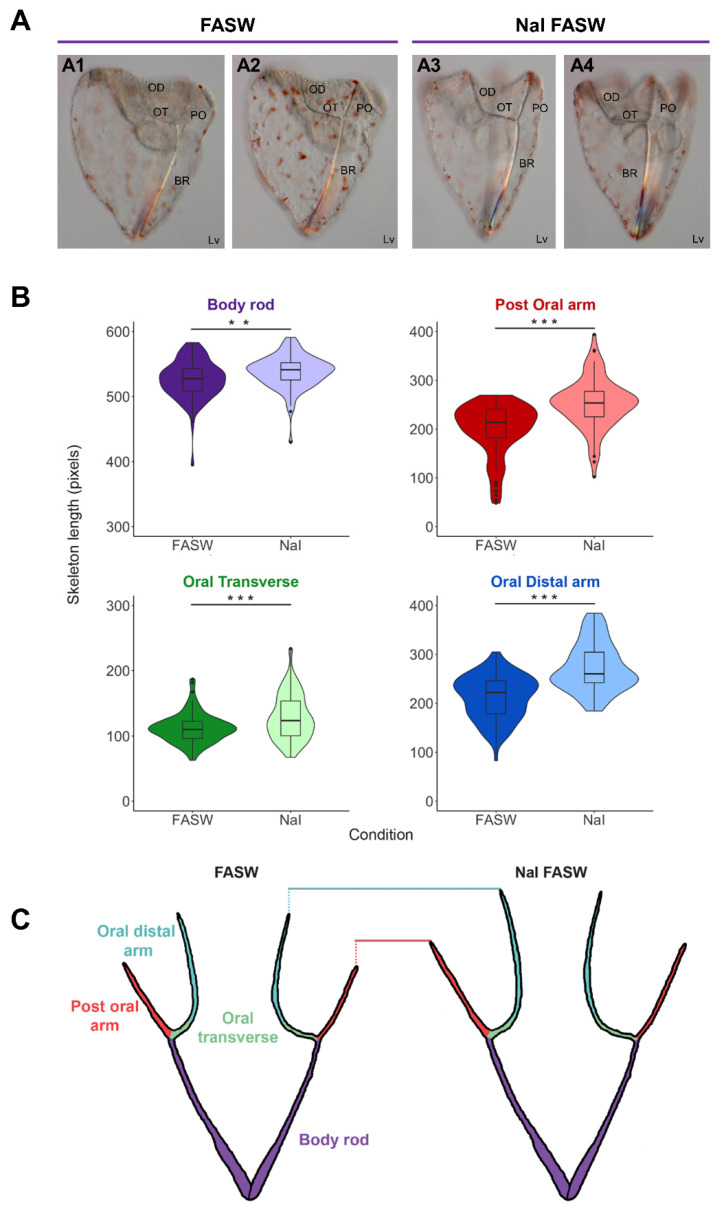
Iodide accelerates skeleton growth in sea urchin larvae. (**A**) The phenotype of 4 dpf *S. purpuratus* pluteus larvae cultured in filtered artificial seawater (**A1**,**A2**) and sodium iodide (NaI) in FASW (**A3**,**A4**), respectively. (**A1**,**A2**) and (**A3**,**A4**) are different focal planes of the same larvae focusing on the right and left arms, respectively. (**B**) Four violin plots with box and whisker plots overlaid plotting the length of the skeleton (pixels) for the different skeletal parts in FASW (*n* = 42) (negative control) and sodium iodide (NaI) in FASW (*n* = 43), measured in larvae collected from three independent experiments. Boxes show the median, lower and upper quartiles, while the whiskers extend to the minimum and maximum data points. A student’s two-tailed t-test was performed, ** (*p* < 0.01) and *** (*p* < 0.001), and reveals that larval skeletal parts are significantly longer when cultured in NaI FASW, with the post-oral, oral transverse, and oral distal skeletal lengths the longest and body rod a little longer. (**C**) Cartoon skeletons illustrating the longer skeletal lengths in the NaI FASW condition compared to normal FASW. BR, Body Rod; OD, Oral Distal; OT; Oral Transverse; PO, Post Oral; Lv, lateral view.

**Figure 3 cells-12-00272-f003:**
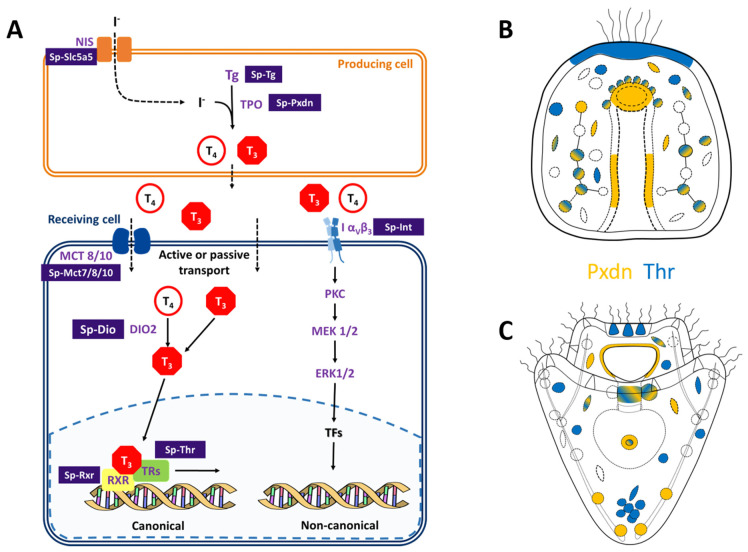
Identification of the THs pathway components in sea urchin genome and identification of putative producing (orange) and/or receiving (blue) cells in sea urchin embryo and pluteus larva. (**A**) Scheme summarizing the data reported in this study. The scheme includes both the canonical and non-canonical integrin-mediated transduction cascades activated by the THs. Names in purple indicate genes belonging to the vertebrate pathway that have homologs in the sea urchin *S. purpuratus* genome. Purple boxes contain the proposed names for the sea urchin homologs of the genes involved in the main TH pathway. (**B**,**C**) Schematic representation of the expression patterns of *Sp-Pxdn* (orange), indicative of a producing cell and *Sp-Thr* (blue), indicative of a receiving cell. at gastrula (**B**) and pluteus (**C**) stages. Color-code is the same as shown in (**A**).

**Table 1 cells-12-00272-t001:** Molecular components of the THs pathway in humans and *S. purpuratus*. Summary of vertebrate genes involved in the TH pathway, including a brief description of their function and homologs identified in the sea urchin genome, the proposed names, and their IDs in the old and new Echinobase versions. Notes on available literature are reported.

*Homo sapiens*		*Strongylocentrotus purpuratus*				
Name	Function	Proposed Name	SPU	WHL	Echinobase Gene Symbol	Notes
**Sodium/iodide symporter** **(NIS)** NP_000444.1	It is a transmembrane glycoprotein that mediates the uptake of iodide in the thyroid gland. It is the first step in the synthesis of thyroid hormones.	*Sp-Slc5a5*	SPU_028043 SPU_028042	WHL22.21242	LOC576134	It has been shown that iodine uptake in sea urchin larvae does not depend on NIS [16,38]
**Thyroglobulin (Tg)** NG_015832.1	It is a glycoprotein that functions as the main precursor for THs and is characterized by several Thyroglobulin type-1 repeats (TY). Its tyrosine residues are combined with iodine and subsequently cleaved to make T4 and T3.	*Sp-Hypp_1359 Sp-Hypp_1727*	SPU_001955 SPU_007546	WHL22.318807 WHL22.616396	LOC755632	XP 001202473 published by [39]. BLAST search on the new version of Echinobase gives this result. The protein encodes for TY repeats
		*Sp-Tg*	SPU_011634	NA	LOC589650	Results obtained through the “search by name” function in Echinobase
**Thyroid peroxidase (TPO)** NP_000538.3	Thyroid peroxidase is an enzyme that catalyzes the oxidative reaction that adds iodine atoms to tyrosine residues of thyroglobulin. This reaction produces the THs T4 and T3	*Sp-Pxdn*	SPU_013889 NA	WHL22.6855 WHL22.6892	LOC593243	It has been shown that a TPO activity is necessary for iodine uptake in the sea urchin *L. variegatus* [16]
**Transthyretin** **(TTR)** NP_000362.1	TTR is a transport protein that mediates the transport of T4 into the bloodstream and cerebrospinal fluid	*Sp-Ttrl*	SPU_011523	WHL22.679950	LOC589020	
**Albumin (ALB)**	Albumin is a family of globular proteins which are responsible for transporting various ligands in the bloodstream.	None				A sea urchin protein known as Endo16, which function is unknown, has been suggested to have originated from an ancestral albumin gene through a duplication event [4,40]
**Thyroxine-binding globulin (TBG)**	TBG is another carrier of THs in the circulation	None				
**Monocarboxylate transporter** NP_006508.2	MCTs are a family of plasma membrane transporters. Also known as SLC16A1- SLC16A14.	*Sp-Mct 7/8/10*	SPU_003433 SPU_020162	WHL22.128984 WHL22.60018	LOC576836	
**Iodothyronine deiodinases** **(DIO)** NP_054644.1 DIO is a subfamily of deiodinase enzymes important in the activation and deactivation of THs. In vertebrates, there are three types of DIOs: Dio1, Dio2 and Dio3		*Sp-Dio*	SPU_002251	WHL22.24216	LOC577015	
		*Sp-Dio1L*	SPU_014348	NA	NA	
		None	SPU_027571	NA	NA	
		None	SPU_015790	NA	NA	
		None	SPU_014347	NA	LOC100893585	
		Thyroxine 5-deiodinase-like	SPU_002251	NA	NA	
		None	NA	NA	LOC100893655	
		None	NA	NA	LOC105440702	
		None	NA	NA	LOC577015	
		Thyroxine 5-deiodinase-like	NA	NA	LOC115927916	
		Type I iodothyronine deiodinase-like	NA	NA	LOC115928194	
		Type I iodothyronine deiodinase-like	NA	NA	LOC100893862	
		Type I iodothyronine deiodinase-like	NA	NA	LOC105442478	
**Thyroid Hormone Receptors** **(TRs)**	TRs are nuclear receptors. Vertebrates have two copies: a TRα and TRβ	*Sp-Thr*	SPU_025239 SPU_018861	WHL22.211956 WHL22.290254	LOC584535	Sea urchins have only one TR. [41] performed a phylogenetic analysis confirming their homology with the human TRs NB Echinobase: SPU_01886 and SPU_025239
**Retinoid X receptor (RXR)**	It is a nuclear receptor. Heterodimerization of RXR with TRs activates gene transcription	*Sp-Rxr*	SPU_028422	WHL22.717794	LOC579018	[41]
**Integrin α_V_β_3_** NP_001138471.2	It is a type of integrin that has been shown to mediate non-genomic response to THs in vertebrates	*Sp-Integrin αP*	SPU_009788; SPU_013535; SPU_013536	WHL22.604820	LOC373206	

## Data Availability

Single cell sequencing data (mapped reads) have been deposited in Dyrad under the unique identifier https://doi.org/10.5061/dryad.n5tb2rbvz, accessed between July–September 2021.

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
