# Peer review of "Molecular and Cellular Characterization of the TH Pathway in the Sea Urchin Strongylocentrotus purpuratus"

_cells, 2023, doi:10.3390/cells12020272_

Round 1
Reviewer 1 Report
Cocurullo and colleagues address an interesting question: is there a an intact thyroid hormone synthesis pathway, as well as a communication axis by which cells receive and interpret this signal, in place in the sea urchin? This manuscript begins with an extensive and thoughtful introduction. They perform an analysis of published sequence information to identify potential orthologs of genes important for TH signaling. Combining published scRNAseq data with FISH, they report the expression patterns for these factors. Last, they include a functional experiment that suggests availability of inorganic iodine influences the development of the sea urchin skeleton, which expresses some of the TH machinery. Overall, I found that this work addresses an interesting question and its conclusions are supported by the data. I support publication but offer a few minor suggestions.
1. In the cases where orthology is ambiguous, could the authors include some structural information? (e.g. “LOC373206 seems to correspond…” line 207) For instance, the integrins are a large family of proteins involved in adhesion and ECM interaction. What domains within this integrin suggest that this is the molecule involved in noncanonical TH signaling? Perhaps some sequence alignments or schematics of domain organization could help. The same analysis could help with Tg, which the authors state does not have an obvious ortholog in the Sp genome (line 198).
2. The experiment in which embryos are cultured +/- iodine is interesting and carefully controlled. But, the results appear to be subtle. First, could the authors please provide representative images of the plutei that were quantified? Do the plutei otherwise look morphologically normal? Second, could the violin plots be labelled with the mean values? Last, could the authors discuss more the functional importance of this finding? Many labs routinely culture embryos in artificial sea water without iodine and achieve fine results. What does this skeletal length difference mean biologically, and how might it be related to physiological development in the ocean? At what regulatory level is skeletogenesis being affected? There is a substantial literature on developmental plasticity in the sea urchin skeleton based on food availability (for instance, Adams et al 2011, Nat Com; Hart & Strathmann 1994, Biol Bull). Could these findings be related and integrated?
3. The FISH images in figure 1 are nice, but I found the DAPI signal to be a little hard to see in the merges. Could this channel be scaled for higher contrast?
4. Could the authors integrate their results more into more discussion/speculation on cell type evolution and the evolutionary origins of a discrete, TH producing organ?
5. The final summary figure is helpful, but could the authors incorporate some of the spatial information into it? Perhaps a summary cartoon of an embryo in which the “producing” and “receiving” cells are differentially highlighted. I understand that there appears to be overlap between these two populations, but this is worth depicting.
6. (minor) Line 76: I thought lamprey were vertebrates?
7. (minor) I suggest using Dorsal-Ventral terminology rather than Oral-Aboral, the latter of which is more of a peculiarity of our echinoderm world and less generalizable to a broader readership.
Author Response
Dear Editor,
We would like to start by thanking all three reviewers for their insightful feedback and their constructive comments that resulted in a substantial improvement of the manuscript.
ANSWER TO REVIEWER 1
- In the cases where orthology is ambiguous, could the authors include some structural information? (e.g. “LOC373206 seems to correspond…” line 207) For instance, the integrins are a large family of proteins involved in adhesion and ECM interaction. What domains within this integrin suggest that this is the molecule involved in noncanonical TH signaling? Perhaps some sequence alignments or schematics of domain organization could help. The same analysis could help with Tg, which the authors state does not have an obvious ortholog in the Sp genome (line 198).
We thank the reviewer for this comment, which allowed us to discover an annotation issue on the integrin genes, for which we apologize. After reanalyzing our data, we concluded that the integrin gene suggested by Taylor and Heyland (ref 31) as mediator of the TH regulation of skeletal growth, Sp-Integrin αP, is present in our single cell transcriptomic dataset and is expressed in the cell types of interest, the skeletal cells; therefore, we excluded the rest of the integrins from our analysis and model. We changed Table 1, figure S1, S2 and S3 and text (lines 215-219) accordingly. About the LOC755632 (putative Tg), a structural analysis has been indeed performed by Belkadi and colleagues (ref 39), and allowed the identification of Thyroglobulin type-1 repeats (TY). We have mentioned this in the text and referenced to their paper (now lines 205-210).
- The experiment in which embryos are cultured +/- iodine is interesting and carefully controlled. But, the results appear to be subtle. First, could the authors please provide representative images of the plutei that were quantified? Do the plutei otherwise look morphologically normal? Second, could the violin plots be labelled with the mean values? Last, could the authors discuss more the functional importance of this finding? Many labs routinely culture embryos in artificial sea water without iodine and achieve fine results. What does this skeletal length difference mean biologically, and how might it be related to physiological development in the ocean? At what regulatory level is skeletogenesis being affected? There is a substantial literature on developmental plasticity in the sea urchin skeleton based on food availability (for instance, Adams et al 2011, Nat Com; Hart & Strathmann 1994, Biol Bull). Could these findings be related and integrated?
We thank the reviewer for these comments. We now included representative pictures of the larvae measured showing normal morphological features in all cell types apart from the length of skeleton (Figure 2 new panel A). Concerning the plots depicted in the new panel C of the same Figure, it is important to note that they already contain a box-plot to highlight median and quartiles, to our opinion representing better the distribution of the single values and the global trend. About the last point, concerning developmental plasticity, our data show that growing larvae in ASW low in iodine does not appear to affect animal general development, as pointed out by the fact that larvae are morphologically normal and display a fully differentiated pigment cells, tripartite gut, and other features typical of this stage of development. Moreover, as the reviewer pointed out, arm length is an important factor for sea urchin larvae in response to food availability, something which we have better explained in the revised manuscript (see lines 468-487). Regarding the level in which this mechanism is affected, we believe it is related to skeletal elongation and growth since morphologically skeletogenic cells appear to be specified and differentiated normally up to pluteus stage regardless of iodine. Overall, by modifying Fig. 2 and the text in regards to these results, we believe we have better integrated in this version of the manuscript our findings on the role of iodine with the current literature on sea urchin phenotypic plasticity.
- The FISH images in figure 1 are nice, but I found the DAPI signal to be a little hard to see in the merges. Could this channel be scaled for higher contrast?
We apologize for this. DAPI fluorescence has been enhanced in all panels.
- Could the authors integrate their results more into more discussion/speculation on cell type evolution and the evolutionary origins of a discrete, TH producing organ?
We appreciate the reviewer’s comment and indeed it would be really interesting to further discuss the evolution of the TH producing organ. Unfortunately, with the current data we believe that this would be too speculative.
- The final summary figure is helpful, but could the authors incorporate some of the spatial information into it? Perhaps a summary cartoon of an embryo in which the “producing” and “receiving” cells are differentially highlighted. I understand that there appears to be overlap between these two populations, but this is worth depicting.
We thank the reviewer for this comment, that made us aware of the lack of a summary figure that is indeed helpful. Since the Pxdn and Thr expression profiles might recapitulate the producing and responding cells respectively, we combined these two patterns in the final scheme (Figure 3 additional panels B and C) to highlight the producing and receiving cells at 2 and 3 dpf.
- (minor) Line 76: I thought lamprey were vertebrates?
We thank the reviewer to spot this. The sentence was rephrased accordingly.
- (minor) I suggest using Dorsal-Ventral terminology rather than Oral-Aboral, the latter of which is more of a peculiarity of our echinoderm world and less generalizable to a broader readership.
Unfortunately, the terms Dorsal-Ventral and Oral-Aboral have been ubiquitously misused in the echinoderm field, especially when referring to the ectoderm. This is also due to the peculiar pyramidal shape of the sea urchin pluteus larva. In the sea urchin larva, indeed, the term Oral ectoderm specifically refers to the ectodermal area surrounding the mouth and encircled by the ciliary band, while the word Aboral ectoderm refers to the rest of the ectoderm, including the entire anal surface. Therefore, in particular when referring to the ectoderm, as in the present study, the Dorsal/Ventral terminology is not be appropriate.
Reviewer 2 Report
The authors used available genomic and transcriptomics resources to identify sea urchin homologs of the Thyroid Hormone (TH) related genes and determine their spatial expression during sea urchin embryogenesis. They identified homologs of most of the vertebrates’ TH genes and used single cell RNA data to predict their expression in multiple cells types including the skeletogenic lineage. For the genes, Pxdn, DIO and Thr they verified these prediction using whole mount in-situ hybridization. To test if the embryo actually utilizes iodine in its development (iodine organification) they added iodine to the water in a culture of sea urchin embryos and detected an increase in the skeletal size in this condition.
Overall the paper is well written and provides novel and interesting information about the TH genes and their expression in the sea urchin embryo that adds to previous studies of the TH function in this organism. However, the conclusion on the important role of the TH pathway in skeletogenesis is not well supported and should be toned down, based on the authors findings: The TH genes are broadly expressed in different tissues, the co-expression of RXR and THR in the skeletogenic cells is observed only in one time point, the TH pathway is expected to promote cell proliferation and growth in general so the iodide could have affected any tissue or all the tissues, so the skeletogenic growth could be an indirect effect of this treatment. Also the spatial expression data is complex and hard to follow, so the authors should summarize their cell lineage studies in a diagram that shows which cells express which genes at the two studied time points (like they did for the regenerating brittle star in BMP biology 2021). Below are my details comments and suggestions to the authors.
Introduction
The introduction is well written and provides the necessary background to understand the the different components of the TH pathway and their role in vertebrates and outside. Some minor correction:
Line 69 – I think residual should be residue.
Line 71 – the example of the amphibian morphogenesis is not explained at all, so it should either be removed or described in another sentence. I think it should be explained because it is referred to later in the discussion.
Line 72 – “outside vertebrates” should start a new paragraph.
Lines 80-81 – I don’t understand the meaning of these two sentences. The animals that don’t produce TH still have the TH response system? Is there other use for iodine? Not clear.
Results
I think you missed a very relevant integrin alpha v because of its name – Sp-Aij, LOC583516, Sp-Aij-1, LOC105439918. This gene is enriched in the skeletogenic cells at 24hpf in Sp according to Barsi et al Genome research 2014 (you can search the data here: http://tulab.genetics.ac.cn:3939/embryonic_territory/) and according to Massri et al Development 2021 is enriched in the skeletogenic and non-skeletogenic mesoderm (you need to look at their data for it). It is also downregulated in VEGFR inhibition (Morgulis et al PNAS 2019, dataset SO2, search for this gene by its name or its WHL22.180881). I am sure you can find it in your single-cell RNA seq and I believe it could be the integrin you’re looking for.
Fig. 3 – since iodine addition could affect many cells in the embryo and not just the skeletogenic cells, it is critical to present images of representative embryos and not only the illustration of the spicules. This way the reader can see the overall morphology of the embryos in this condition.
Lines 345-348: As mentioned before I don’t agree that this study demonstrated the important role of iodide in skeletogenesis – due to the broad expression of the genes, since THR and RXR are only co-expressed in the skeletogenic cells at one time point and since the iodine treatment could affect all the tissues. Please explain that the results you see could be through direct and indirect effect on skeletogenesis.
I wish you had made functional studies of at least one gene, but I guess it is outside the scope of this paper.
Discussion –
Lines 280 - “out of the four DIOs in our analysis” and 387-390 -“the existence of many Sp-DIOs” – in the paper, table, sup and figures there is only one DIO gene mentioned. If there are four genes they should be listed in the table like in the case of the other genes. If there is only one gene, the statements about the many genes should be corrected.
Line 392 – “Finally, the existence of Sp-Thr and Sp-Rxr as well as integrins, suggests that both canonical and non-canonical transduction cascades in response to THs might occur in sea urchins.“ Integrins play critical role in cell adhesion and other signaling pathway. so their presence is not an indication for the TH function/non-canonical pathway. Please remove “integrins” from this sentence, or if you can, be more specific.
Line 406 – “while the expression” should start a new paragraph as it is unrelated to the previous sentences.
This is practically impossible to understand which cells are expressing which TH genes at the two studied time points. For example - you say that DIO is expressed in the skeletogenic cells so those are the ones making TH, and then you say that its produced in the aboral ectoderm and suggest paracrine effect. It is confusing. Please draw an image of the embryo at 2dpf and 3dpf where you plot the predicted/verified expression of all the genes you tested so it will be obvious which genes are co-expressed where and when. It is critical to the interpretation of your results.
Author Response
Dear Editor,
We would like to start by thanking all three reviewers for their insightful feedback and their constructive comments that resulted in a substantial improvement of the manuscript.
ANSWER TO REVIEWER 2
- Line 69 – I think residual should be residue.
Residual has been replaced by the word residue
- Line 71 – the example of the amphibian morphogenesis is not explained at all, so it should either be removed or described in another sentence. I think it should be explained because it is referred to later in the discussion.
As pointed out by the reviewer, this point was important for the discussion, therefore we decided to further explain it. See lines (72-73)
- Line 72 – “outside vertebrates” should start a new paragraph.
We agree with the reviewer and fixed this point.
- Lines 80-81 – I don’t understand the meaning of these two sentences. The animals that don’t produce TH still have the TH response system? Is there other use for iodine? Not clear.
We apologize for not being clear on this point. In general, it has been shown that animals lacking a mechanism for TH production, can still respond to externally provided THs (e.g. from food source), through either TH receptor or different mechanisms. For example, in the case of tunicates, reported in reference 10, even though THs appear to accelerate metamorphosis and settlement, there is no real evidence for their synthesis nor for the existence of a functional receptor in these animals. We modified the text in lines 75-89 to better explain these cases.
- I think you missed a very relevant integrin alpha v because of its name – Sp-Aij, LOC583516, Sp-Aij-1, LOC105439918. This gene is enriched in the skeletogenic cells at 24hpf in Sp according to Barsi et al Genome research 2014 (you can search the data here: http://tulab.genetics.ac.cn:3939/embryonic_territory/) and according to Massri et al Development 2021 is enriched in the skeletogenic and non-skeletogenic mesoderm (you need to look at their data for it). It is also downregulated in VEGFR inhibition (Morgulis et al PNAS 2019, dataset SO2, search for this gene by its name or its WHL22.180881). I am sure you can find it in your single-cell RNA seq and I believe it could be the integrin you’re looking for.
We thank a lot the reviewer for this comment that, together with the one from Reviewer 1, made us go back to our integrin analysis, and resolving an annotation issue concerning intergrin alpha P. Moreover, since the integrin Sp-Aij was shown to be potentially involved in skeletogenesis in other sea urchin species and to be expressed at earlier stages during S. pupuratus embryogenesis we followed the reviewers’ suggestion and looked in our single cell data for that specific gene. Unfortunately, we could not find any evidence for its expression in our scRNA-seq data, suggesting that either it is not expressed at 48 and 72 hpf or that its expression levels are too low. We modified Table 1, figure S1, S2 and S3 and text (lines 215-219) according to this updated analysis.
- 3 – since iodine addition could affect many cells in the embryo and not just the skeletogenic cells, it is critical to present images of representative embryos and not only the illustration of the spicules. This way the reader can see the overall morphology of the embryos in this condition.
We thank the reviewer for their comments and suggestion to include representative pictures of the larvae measured to show overall morphology of the larvae. As mentioned already in response to reviewer 1 (point 2) we included in Figure 2 a new panel A showing normal morphological features in all cell types apart from the skeleton length.
- Lines 345-348: As mentioned before I don’t agree that this study demonstrated the important role of iodide in skeletogenesis – due to the broad expression of the genes, since THR and RXR are only co-expressed in the skeletogenic cells at one time point and since the iodine treatment could affect all the tissues. Please explain that the results you see could be through direct and indirect effect on skeletogenesis.
We appreciate the constructive criticism of the reviewer, although we do not fully agree on this point. Based on the available literature, and especially the paper by Taylor and Heyland (ref 31), it was clearly demonstrated that THs control skeletal formation at gastrula stage, the same stage in which we detect the coalescence in skeletal cells of genes involved both in the canonical (Thr-Rxr) and non-canonical (Integrin αP) regulation. According to our data, it appears that at the stage analyzed (at least at a morphological level) only skeletal growth deficits are observed, suggesting a clear role of iodine in skeletal development. The fact that the genes involved in this are co-expressed in skeletal cells only at gastrula stage, does not exclude the possibility of Iodine controlling skeletal development as suggested by the reviewer, but on the contrary suggests that developmental decisions of the rate of skeletal growth could be taken before the rods appear as distinct structures. Moreover, as shown in new Figure 2 A, the overall phenotype of the larva appears normal, except for the skeletal growth. Of course other metabolic and physiological phenotypes could be present in other cell types, however this is beyond the scope of this study that it is centered on the developmental role of iodine and TH.
- I wish you had made functional studies of at least one gene, but I guess it is outside the scope of this paper.
As the reviewer pointed out, this was beyond the scope of our paper, however, we hope that in the future we could address the role of several genes of the TH pathway and see at which level of the regulatory network skeletal growth is impaired.
- Lines 280 - “out of the four DIOs in our analysis” and 387-390 -“the existence of many Sp-DIOs” – in the paper, table, sup and figures there is only one DIO gene mentioned. If there are four genes they should be listed in the table like in the case of the other genes. If there is only one gene, the statements about the many genes should be corrected.
We thank the reviewer for this comment. We included the complete list of DIOs identified in the S. purpuratus genome in Table 1.
- Line 392 – “Finally, the existence of Sp-Thr and Sp-Rxr as well as integrins, suggests that both canonical and non-canonical transduction cascades in response to THs might occur in sea urchins.“ Integrins play critical role in cell adhesion and other signaling pathway. so their presence is not an indication for the TH function/non-canonical pathway. Please remove “integrins” from this sentence, or if you can, be more specific.
We appreciate the reviewer’s comment and we agree that this statement was indeed confusing. In the current version of the manuscript, we have limited our analysis to the specific integrin Sp-Integrin αP, which is indicative of non/canonical pathway (Taylor and Heyland, ref 31), we modified the sentence referring solely to this gene (line 414).
- Line 406 – “while the expression” should start a new paragraph as it is unrelated to the previous sentences.
We apologize for this. It has been corrected.
- This is practically impossible to understand which cells are expressing which TH genes at the two studied time points. For example - you say that DIO is expressed in the skeletogenic cells so those are the ones making TH, and then you say that its produced in the aboral ectoderm and suggest paracrine effect. It is confusing. Please draw an image of the embryo at 2dpf and 3dpf where you plot the predicted/verified expression of all the genes you tested so it will be obvious which genes are co-expressed where and when. It is critical to the interpretation of your results.
We thank a lot the reviewer for this comment and included in the new Figure 3 a final scheme of a 2 dpf and a 3dpf larva in which we highlighted the Pxdn/producing and the Thr/responding cells. We did not include other genes as we thought this would make the final scheme hard to understand.
Reviewer 3 Report
The authors provide new data on the presence and distribution of thyroid hormone signaling, synthesis and transport related transcripts in the sea urchin Strongylocentrotus purpuratus. They also present data on iodine exposure of embryos/larvae that shows that skeletal arms are elongated when larvae are exposed to iodine. The identification and mapping of transcripts using single cell sequencing and fluorescent ISH is informative in the context of work done by other research that shows that THs can accelerate skeletogenesis in embryos, accelerate juvenile development in juveniles, sea urchins can synthesize THs from incorporated iodine, may receive THs and/or iodine from food (unicellular algae), that algae can incorporate iodine and synthesize THs and finally, provide evidence for the presence of an integrin TH receptor. The work is also informative evolutionarily, as it shows in detail, WHERE specific genes are expressed in the embryo and early larvae, allowing potential inference about the TH signaling and synthesis system in echinoderms.
Major issues:
1. Iodine exposure: Unfortunately, the iodine experiment data presented in this manuscript are not very informative. They lack proper controls, which need to be conducted in order to draw specific conclusions about TH signaling. Miller and Heyland (2013 – JEB) provided a detailed analysis on the putative mechanisms of iodine uptake in S. purpuratus eggs and larvae, using radioactive labeled iodine and various antagonists of transport systems. This work provided a model for how iodine is likely incorporated and that it is using a peroxidase facilitated diffusion mechanism, similarly to what was shown in unicellular algae by two research groups (Frontiers | Characterization of Iodine-Related Molecular Processes in the Marine Microalga Tisochrysis lutea (Haptophyta) | Marine Science (frontiersin.org) and Uptake of iodide in the marine haptophyte Isochrysis sp. (T.ISO) driven by iodide oxidation - PubMed (nih.gov). In this work, which should also be cited in the manuscript, the authors clearly outline that most salts used to make artificial seawater (especially NaI) contain iodine at concentrations comparable to what can be found in seawater. Therefore, any addition of iodine in experiments is increasing the iodine concentration above what larvae and embryos are typically exposed to. Furthermore, as outlined in these papers, it is critical to conduct control experiments with pharmacological agents, to block iodine update or antagonize it, in order to test whether the effect is specific to iodo-tyrosines or an indirect effect of excess iodine in the larvae. For example, the authors could and should use some of the several inhibitors that were outlined in Miller and Heyland 2013 (H2O2, reducing agents, peroxidase inhibitors, a metabolic inhibitor and an anion channel inhibitors for sodium independent iodine uptake and potassium perchlorate (KClO4) and/or N-methyl-D-glucamine for sodium dependent uptake. Note also that Miller and Heyland never actually tested gastrula stages and sodium dependent mechanisms should at least be tested in that stage. If the authors find similar arm lengths when NaI is provided in combination with these antagonists they can conclude that iodination of tyrosines or direct iodine effects are in fact responsible for skeletal rod elongation. These controls are important, as Taylor and Heyland 2018 did not find any evidence for arm length elongation by eith erT3 or T4, but rather an acceleration of arm length. It will be critical to properly validate this interesting result the author report on iodine treatment leading to skeleton elongation by conducting proper controls.
2. Physiological experiments: The authors show that TH signaling related transcripts are expressed in specific structures in normally developing larvae and draw specific conclusions about the TH signaling pathway. While some of this data is insightful, it lacks experiments where larvae are stimulated with THs. For example, when THR in mammals is binding T3, one would typically see an up regulation of THR (auto-regulation), which would be very strong evidence for the function of THR in TH signaling. Similarly, integrins, functioning as heterodimers, may be up-regulated by TH treatment and form function TH-receptor complexes, as discussed by Taylor and Heyland 2018. Fundamentally, the distribution of TH related transcripts without stimulation of the pathway or inhibition of TH synthesis in larvae, only tells a small part of the story.
3. The manuscript is missing some important references:
a. Guillaume and Laudet 2016 which discussed thyroid hormone function in annelids and that should be mentioned in the introduction. Generally, the statement that “thyroid hormone synthesis is rare in invertebrates” is just simply not justified – see Taylor and Heyland 2017 and Eales 1994
b. Susan and Lennarz 2000 and especially Marsden and Burke 1997 determined that alphav beta 3 homologs are highly expressed in the gastrula, and localized to the PMCs and an area of the midgut that exactly matches their Pxdn expression the authors report. This has to be incorporated and discussed.
c. The Miller and Heyland work in JEB (2013) needs to be cited and the information needs to be used (as outlined above) as it is the only paper so far that analyzes the iodine uptake and organification mechanism in sea urchins
d. Work on micro-algae (also mentioned above) needs to be cited, as it provides some context on the long evolutionary history of iodine metabolism. Specifically, it suggests that TH synthesis preceded animal evolution
1.
Minor issues:
P1: better rephrase “during urchin development” to read “sea urchin embryogenesis and early larval development”. The authors do not provide any information on later and metamorphic stages
P2: The authors write “… 'mostly mediated by the Thyroid Hormone Receptors (TRs)” this statement neglects to mention that several extensive reviews have pointed out other non-genomic pathways, at least in vertebrates – and that work should eb acknowledged: Davis, Armstrong, Comment: in vertebrates, but alternative pathways have been studied for several decades - Davis, Armstrong, Taylor etc.
P2: The authors write “… 'which residual is involved during the deiodinase reaction” should be changed to residue
P2: The authors write “In fact, most invertebrates cannot produce THs by themselves and rely on external sources of iodinated compounds such as algae (by ingestion)” . This statement is premature. Considering that algae and bacteria likely synthesize THs and evidence was published by others that sea urchins can synthesize THs from incorporated iodine and showed that unfed larvae can settle when exposed to T4. Their work does not exclude the possibility that the hormone can be synthesized endogenously.
P2: The authors state that “… 'Their data suggests that exposure to T4 results in arm growth arrest at early larval stage, while it induces the metamorphosis of competent larvae”. This statement is incorrect and needs to be revised. 1) there is no specific evidence that there is a growth arrest, although this might be a possible explanation for shorter arms. It is also possible that arms just grow slower. I suggest the authors restrict the statement to TH treatment results in shorter arms. 2) there is no evidence to date that TH INDUCES metamorphosis. While it accelerates the development of juvenile structures, it results in earlier settlement and this has been shown in several species of echinoids.
P. 3: The authors state a series of questions, some of which should be revised or removed as they are either not directly addressed by this paper or have been investigated and conclusive evidence has been presented. For example, the authors write “Can sea urchin larvae produce THs?” The answer to that question has been investigated and the answer is yes.
P.9: Typo – should read at the gastrula stage
P. 10: I suggest expanding or at least modifying the discussion of integrin expression. In addition to the comment above, that integrin ISH have been previously conducted, there is the possibility that expression of subunits is regulated by the presence of THs. One important question about integrin TH receptors, is how the process is regulated. One issue is that it is not intuitively clear what the pre-hormone for T4 would be and related to that, we simply don’t know enough about the regulation of integrin subunits. If for example, T4 actiates the generation of the binding complex (heterodimer) that could be a hypothetical explanation about how TH signaling is regulated.
P. 13: Typo – rely instead of relay?
P. 13: For reasons outlined above, the statement “'Importantly , larvae reared in FASW (which lacks iodine) simply cannot stand as is. It is extremely difficult to make iodine free seawater – please also refer to Miller and Heyland 2013.
Author Response
Dear Editor,
We would like to start by thanking all three reviewers for their insightful feedback and their constructive comments that resulted in a substantial improvement of the manuscript.
ANSWER TO REVIEWER 3
- Iodine exposure: Unfortunately, the iodine experiment data presented in this manuscript are not very informative. They lack proper controls, which need to be conducted in order to draw specific conclusions about TH signaling. Miller and Heyland (2013 – JEB) provided a detailed analysis on the putative mechanisms of iodine uptake in S. purpuratus eggs and larvae, using radioactive labeled iodine and various antagonists of transport systems. This work provided a model for how iodine is likely incorporated and that it is using a peroxidase facilitated diffusion mechanism, similarly to what was shown in unicellular algae by two research groups (Frontiers | Characterization of Iodine-Related Molecular Processes in the Marine Microalga Tisochrysis lutea (Haptophyta) | Marine Science (frontiersin.org) and Uptake of iodide in the marine haptophyte Isochrysis sp. (T.ISO) driven by iodide oxidation - PubMed (nih.gov). In this work, which should also be cited in the manuscript, the authors clearly outline that most salts used to make artificial seawater (especially NaI) contain iodine at concentrations comparable to what can be found in seawater. Therefore, any addition of iodine in experiments is increasing the iodine concentration above what larvae and embryos are typically exposed to. Furthermore, as outlined in these papers, it is critical to conduct control experiments with pharmacological agents, to block iodine update or antagonize it, in order to test whether the effect is specific to iodo-tyrosines or an indirect effect of excess iodine in the larvae. For example, the authors could and should use some of the several inhibitors that were outlined in Miller and Heyland 2013 (H2O2, reducing agents, peroxidase inhibitors, a metabolic inhibitor and an anion channel inhibitors for sodium independent iodine uptake and potassium perchlorate (KClO4) and/or N-methyl-D-glucamine for sodium dependent uptake. Note also that Miller and Heyland never actually tested gastrula stages and sodium dependent mechanisms should at least be tested in that stage. If the authors find similar arm lengths when NaI is provided in combination with these antagonists they can conclude that iodination of tyrosines or direct iodine effects are in fact responsible for skeletal rod elongation. These controls are important, as Taylor and Heyland 2018 did not find any evidence for arm length elongation by eith erT3 or T4, but rather an acceleration of arm length. It will be critical to properly validate this interesting result the author report on iodine treatment leading to skeleton elongation by conducting proper controls.
We believe that there was a misunderstanding on this point. The scope of our study is to complement the clear observation that the lack of iodine supplement in the artificial sea water (as per recipe provided in the methods) produces larvae having shorter skeletal rods compared to the one raised in presence of physiological concentration of iodine (see methods) with TH pathway gene expression data. We want to point out that up to 3 dpf the embryos are grown without any food source and in presence of antibiotics to avoid bacterial growth. We further clarify this in the current version of the manuscript referring to ASW poor in iodine and ASW enriched with iodine supplement. We think that our results clearly show that, without iodine supplement, larvae cannot grow as long skeletal rods as they would have if provided with a proper Iodine source. Noteworthy, this effect is visible also on the body rod (BR), confirming that it corresponds to a general impairment of the skeletal growth rather that a secondary physiological response. In fact, invariant BR lengths in S. purpuratus were used by Adams and colleagues (2011) as controls to show the regulatory effect of food on skeletal arm length (which did not affect BR). Moreover, the expression of most of the genes involved in the THs signaling, from production to response, in specific and relevant cell types, strongly support the idea of a specific iodine mechanism to produce THs, rather than a broad, unclear, mechanism of action of iodine alone. This is supported also by the fact that iodine supplement did not induce any other morphological change, to be expected in case the hypothesis for a broad iodine action were true. In conclusion in the revised version of the manuscript by adding larva images in Figure 2 (panel A) and discussing the experiment in regards to the well-known phenotypic plasticity of the sea urchin larva, we clarified the role of iodine in skeletal growth (see lines 467-485). Finally, these data are in agreement with Taylor and Heyland 2018 studies and complement them.
- Physiological experiments: The authors show that TH signaling related transcripts are expressed in specific structures in normally developing larvae and draw specific conclusions about the TH signaling pathway. While some of this data is insightful, it lacks experiments where larvae are stimulated with THs. For example, when THR in mammals is binding T3, one would typically see an up regulation of THR (auto-regulation), which would be very strong evidence for the function of THR in TH signaling. Similarly, integrins, functioning as heterodimers, may be up-regulated by TH treatment and form function TH-receptor complexes, as discussed by Taylor and Heyland 2018. Fundamentally, the distribution of TH related transcripts without stimulation of the pathway or inhibition of TH synthesis in larvae, only tells a small part of the story.
We agree with the reviewer on the fact that we address a small part of the story; however, we think that investigating such a fascinating and enigmatic signaling is not work for a single limiting paper. Some of the questions raised by the reviewer have been already analyzed and discussed by Heyland and collaborators and it would be superfluous to redo the same experiments. On the other side this study complements Heyland’s lab findings addressing the important question of capacity of the sea urchin larva to sustain an organification of iodine, given the condition in which the larvae are raised and the presence of the TH pathway genetic toolkit. On the contrary, we are convinced that to fully address all the remaining mysteries (such as the existence of the Tg, the function of Pxdn in the THs synthesis and of the other genes identified) it deserves dedicated works with functional studies on several genes and genome wide analyses. Indeed, all this was beyond the scope of our manuscript, which focuses on identifying the possible components of the pathway, investigating their expression profiles at two relevant embryonic developmental stages, and finally investigating the biological effect of lack of iodine on skeletal growth. Moreover, considering the fact that larvae were grown without food and that at gastrula stage larvae do not eat, the “ingestion source hypothesis” seems quite unlikely.
- The manuscript is missing some important references:
- Guillaume and Laudet 2016 which discussed thyroid hormone function in annelids and that should be mentioned in the introduction. Generally, the statement that “thyroid hormone synthesis is rare in invertebrates” is just simply not justified – see Taylor and Heyland 2017 and Eales 1994
- Susan and Lennarz 2000 and especially Marsden and Burke 1997 determined that alphav beta 3 homologs are highly expressed in the gastrula, and localized to the PMCs and an area of the midgut that exactly matches their Pxdn expression the authors report. This has to be incorporated and discussed.
- The Miller and Heyland work in JEB (2013) needs to be cited and the information needs to be used (as outlined above) as it is the only paper so far that analyzes the iodine uptake and organification mechanism in sea urchins
- Work on micro-algae (also mentioned above) needs to be cited, as it provides some context on the long evolutionary history of iodine metabolism. Specifically, it suggests that TH synthesis preceded animal evolution
We appreciate the reviewer’s suggestions to include the aforementioned references. We have updated the manuscript accordingly. More in details:
- Guillaume and Laudet 2016 was added in lines 78-79 along with the following description “Nonetheless, thyroglobulin was suggested to be a vertebrate novelty, therefore the mechanism by which THs are produced outside this taxon is still a mystery [23]”
- Susan and Lennarz 2000 was cited in lines 216-219
“Moreover, Susan and collaborator in 2000 [42] found this integrin to be expressed in S. purpuratus at low levels at early embryonic stages and at higher levels from gastrula stage, with the highest expression at pluteus larva stage.”
- Miller and Heyland work in JEB (2013) was cited in lines 96-97
“A peroxidase-dependent mechanism for the iodine uptake was subsequently found also in the species Strongylocentrotus purpuratus [30]”
- Heidelberg et all, 2018, was incorporated to the preexisting sentence referring to algae (micro and macro) being able to produce THs in lines 75-78
“Despite the fact that the thyroid gland as a distinct organ is a novelty of vertebrates, it has been shown that other chordates [6,15], some invertebrates [16], as well as non-animal representatives such as algae [17–19], that lack a proper thyroid organ, are also able to produce and/or to respond to THs (for reviews see [20–22]).”
- P1: better rephrase “during urchin development” to read “sea urchin embryogenesis and early larval development”. The authors do not provide any information on later and metamorphic stages.
We thank the reviewer. The sentence has been rephrased.
- P2: The authors write “… 'mostly mediated by the Thyroid Hormone Receptors (TRs)” this statement neglects to mention that several extensive reviews have pointed out other non-genomic pathways, at least in vertebrates – and that work should eb acknowledged: Davis, Armstrong, Comment: in vertebrates, but alternative pathways have been studied for several decades - Davis, Armstrong, Taylor etc.
We think there is a misunderstanding here. We also agree that the non-canonical pathways are of great importance. In fact, we had already mentioned their presence and function in the original version of the manuscript (Lines 58-63) and had already included and acknowledged several relevant works:
- Davis, P.J.; Goglia, F.; Leonard, J.L. Nongenomic actions of thyroid hormone. Nat. Rev. Endocrinol. 2016, 12, 111–121, doi:10.1038/nrendo.2015.205.
- Davis, F.B.; Mousa, S.A.; O’Connor, L.; Mohamed, S.; Lin, H.-Y.; Cao, H.J.; Davis, P.J. Proangiogenic Action of Thyroid Hormone Is Fibroblast Growth Factor–Dependent and Is Initiated at the Cell Surface. Circ. Res. 2004, 94, 1500–1506, doi:10.1161/01.RES.0000130784.90237.4A.
- Liu, X.; Zheng, N.; Shi, Y.-N.; Yuan, J.; Li, L. Thyroid hormone induced angiogenesis through the integrin αvβ3/protein kinase D/histone deacetylase 5 signaling pathway. J. Mol. Endocrinol. 2014, 52, 245–254, doi:10.1530/JME-13-0252.
However, we agree with this reviewer on the fact that this sentence 'mostly mediated by the Thyroid Hormone Receptors (TRs)” is misleading and therefore we toned down the wording by removing the word mostly.
- P2: The authors write “… 'which residual is involved during the deiodinase reaction” should be changed to residue
We thank the reviewer for pointing out this mistake. We fixed it.
- P2: The authors write “In fact, most invertebrates cannot produce THs by themselves and rely on external sources of iodinated compounds such as algae (by ingestion)” . This statement is premature. Considering that algae and bacteria likely synthesize THs and evidence was published by others that sea urchins can synthesize THs from incorporated iodine and showed that unfed larvae can settle when exposed to T4. Their work does not exclude the possibility that the hormone can be synthesized endogenously.
We believe this to be a result of confusion. As we explained in the introduction, the fact that sea urchin can synthesize THs, was never actually proved. There is evidence for the iodine uptake though, which we mentioned. Moreover, the fact that “most invertebrates cannot produce THs by themselves and rely on external sources of iodinated compounds such as algae (by ingestion)” is not a result of the interpretation of our data, but only a general statement coming from relevant literature in the field, as described by the works of Holzer, Roux and Laudet (ref 10) or Heyland and Moroz (ref 27), which we mention on this point (lines 86-88).
- P2: The authors state that “… 'Their data suggests that exposure to T4 results in arm growth arrest at early larval stage, while it induces the metamorphosis of competent larvae”. This statement is incorrect and needs to be revised. 1) there is no specific evidence that there is a growth arrest, although this might be a possible explanation for shorter arms. It is also possible that arms just grow slower. I suggest the authors restrict the statement to TH treatment results in shorter arms. 2) there is no evidence to date that TH INDUCES metamorphosis. While it accelerates the development of juvenile structures, it results in earlier settlement and this has been shown in several species of echinoids.
We thank the reviewer for pointing out this misunderstanding and we fixed the sentence in line 97-98 accordingly.
- 9: Typo – should read at the gastrula stage
The typo has been corrected.
- 3: The authors state a series of questions, some of which should be revised or removed as they are either not directly addressed by this paper or have been investigated and conclusive evidence has been presented. For example, the authors write “Can sea urchin larvae produce THs?” The answer to that question has been investigated and the answer is yes.
We appreciate the reviewer’s comment however, we do not agree that this point was already investigated and fully answered. Although our data contributed to a better understanding of TH regulation and its role of larval growth in sea urchins, we believe that this specific question “Can sea urchin larvae produce THs?” still remains partially unanswered.
- 10: I suggest expanding or at least modifying the discussion of integrin expression. In addition to the comment above, that integrin ISH have been previously conducted, there is the possibility that expression of subunits is regulated by the presence of THs. One important question about integrin TH receptors, is how the process is regulated. One issue is that it is not intuitively clear what the pre-hormone for T4 would be and related to that, we simply don’t know enough about the regulation of integrin subunits. If for example, T4 actiates the generation of the binding complex (heterodimer) that could be a hypothetical explanation about how TH signaling is regulated.
As mentioned above, replying to reviewer 1 and 2, our previous analysis of integrin expression in the sea urchin larva was affected by an annotation issue. Once solved this issue, our results in the present version of the manuscript are in line with what shown by previous studies by Taylor and Heyland (ref 31). However, while the questions posed by this reviewer are of great interest, we believe that our present analysis cannot offer any further speculation on this issue.
- 13: Typo – rely instead of relay?
We thank the reviewer for pointing out this mistake. It has been fixed.
- For reasons outlined above, the statement “'Importantly, larvae reared in FASW (which lacks iodine) simply cannot stand as is. It is extremely difficult to make iodine free seawater – please also refer to Miller and Heyland 2013.
As explained above, we understand that the scope of the experiment was not clear. The aim was never to make specifically free-iodine water, but to show that a low iodine content can have important biological effects. We therefore modified the text throughout the manuscript accordingly.
Round 2
Reviewer 2 Report
The authors addressed all my comments in an appropriate way. I didn't go over the text carefully to check for spelling and typos - I leave it for the authors and for the proofs. Nice work!
Author Response
The authors addressed all my comments in an appropriate way. I didn't go over the text carefully to check for spelling and typos - I leave it for the authors and for the proofs. Nice work!
We thank the reviewer for the nice comment.
Reviewer 3 Report
The authors addressed many of the issue raised. However, two major issues remain:
1. Iodine concentration:
Average iodine concentration in seawater is around 50ug/L but can also be highly variable depending on the region. The authors decided to add 57ug/L for their iodine treatment. NaCl, one of the main components of their artificial seawater has a maximum of 0.002% residual iodine in it (Sodium chloride, 99+%, ACS reagent, Thermo Scientific™ (thermofisher.com)). Based on my calculation this corresponds to 566ug/L (0.002% of 28.3g) of iodine based on just the NaCl added to seawater, which is 10 times more than the excess added by the treatment. In other words, the authors are increasing the iodine concentration in their treatment by 10% above ambient. As mentioned previously, I don't think this is sufficient to draw the conclusion that iodine treatment is the cause for body skeleton elongation. The amount (and molarity) of iodine present in seawater is more than sufficient to synthesize THs. It also contradicts the statement in the author's response that seawater is poor in iodine. Their treatment only adds a fraction of what is already present. If there are doubts about the actual amount of iodine in seawater, the authors should at least measure iodine in their ASW and report the data to ensure that their treatment actually is a significant increase above ambient. Either way, their explanation of BR elongation should also be discussed in the context of PMC's contribution to BR elongation, reported by Ettensohn and Malinda in Development (1993). Their work comes to the conclusion that "We find that the rate of skeletal rod elongation is independent of both the mode of rod growth (chain or plug) and the number of PMCs in the plug at the growing rod tip. Instead, the rate of elongation appears to be strictly regulated by the quantity of ectodermal tissue present in the embryo." The work presented here makes the assumption, from my understanding, that PMCs are providing the THs for skeletal extension. This also is somewhat in conflict with work published by Taylor and Heyland 2018 that suggest that it is the timing that is impacted by TH treatment and not the extension, further emphasizing the importance that an experiment manipulating iodine as a treatment has to provide data on iodine removal as a control. I simply do not feel comfortable with the lack of controls in this experiment in light of the fact that the amount of iodine added is minor compared to what is present in seawater.
2. TH synthesis
In their response to my review, the authors state that "As we explained in the introduction, the fact that sea urchin can synthesize THs, was never actually proved". This is incorrect. We showed in Heyland et al. 2006 (JEZ-B) - Figure 2C that T4 (not T3) is present in response to I125 exposure of sea urchin larvae and that the treatment of these larvae with thiourea leads to the absence of radioactive T4. While the authors may consider the evidence by TLC as insufficient, this is evidence towards the synthesis of THs from incorporated iodine and should be discussed. Therefore, stating that it has not been proven is misleading.
Author Response
Reviewer: The authors addressed many of the issue raised. However, two major issues remain:
- Iodine concentration:
Average iodine concentration in seawater is around 50ug/L but can also be highly variable depending on the region. The authors decided to add 57ug/L for their iodine treatment. NaCl, one of the main components of their artificial seawater has a maximum of 0.002% residual iodine in it (Sodium chloride, 99+%, ACS reagent, Thermo Scientific™ (thermofisher.com)). Based on my calculation this corresponds to 566ug/L (0.002% of 28.3g) of iodine based on just the NaCl added to seawater, which is 10 times more than the excess added by the treatment. In other words, the authors are increasing the iodine concentration in their treatment by 10% above ambient. As mentioned previously, I don't think this is sufficient to draw the conclusion that iodine treatment is the cause for body skeleton elongation. The amount (and molarity) of iodine present in seawater is more than sufficient to synthesize THs. It also contradicts the statement in the author's response that seawater is poor in iodine. Their treatment only adds a fraction of what is already present. If there are doubts about the actual amount of iodine in seawater, the authors should at least measure iodine in their ASW and report the data to ensure that their treatment actually is a significant increase above ambient. Either way, their explanation of BR elongation should also be discussed in the context of PMC's contribution to BR elongation, reported by Ettensohn and Malinda in Development (1993). Their work comes to the conclusion that "We find that the rate of skeletal rod elongation is independent of both the mode of rod growth (chain or plug) and the number of PMCs in the plug at the growing rod tip. Instead, the rate of elongation appears to be strictly regulated by the quantity of ectodermal tissue present in the embryo." The work presented here makes the assumption, from my understanding, that PMCs are providing the THs for skeletal extension. This also is somewhat in conflict with work published by Taylor and Heyland 2018 that suggest that it is the timing that is impacted by TH treatment and not the extension, further emphasizing the importance that an experiment manipulating iodine as a treatment has to provide data on iodine removal as a control. I simply do not feel comfortable with the lack of controls in this experiment in light of the fact that the amount of iodine added is minor compared to what is present in seawater.
Our response:
About the first point, we do not agree with the reviewer. In our experiments we bred sea urchin embryos until larval stage in artificial sea water without (control condition) and with Sodium Iodide at a final concentration of 57 μg/L. Subsequently, we evaluated the phenotype of larvae reared in the same medium without and with the additional source of iodine in order to understand if and how iodine was necessary for the larval growth. We found that larvae reared in ASW with the addition of extra Sodium Iodide have statistically significant longer skeletal rods, indicating that iodine is indeed involved in the skeleton elongation. We acknowledge the existence of iodine traces in the salts commercially available, however the traces present are not exactly known, only declared as <0.002%. Note that in our experiments the very same solution of ASW was used to rear larvae in both control and treatment conditions and the treatment condition consist in adding Sodium Iodide as additional source of iodine. This protocol is the appropriate procedure to evaluate the effect/function of a substance. The existence of iodine traces in the starting material, therefore, is not relevant to our results. Finally, while we agree that addressing the effect of such iodine traces might be interesting because many groups lacking of a natural source of sea water uses ASW to grow their specimens, addressing this topic is completely out of the scope of our paper.
Reviewer:
- TH synthesis
In their response to my review, the authors state that "As we explained in the introduction, the fact that sea urchin can synthesize THs, was never actually proved". This is incorrect. We showed in Heyland et al. 2006 (JEZ-B) - Figure 2C that T4 (not T3) is present in response to I125 exposure of sea urchin larvae and that the treatment of these larvae with thiourea leads to the absence of radioactive T4. While the authors may consider the evidence by TLC as insufficient, this is evidence towards the synthesis of THs from incorporated iodine and should be discussed. Therefore, stating that it has not been proven is misleading.
Our response
About the second point, again we do not agree with the reviewer. It is widespread in the literature the notion that most commonly echinoderms are known to not produce THs. Moreover, Heyland and collaborators in the studies mentioned by the reviewer were focusing on a different sea urchin species (L. variegatus) and therefore their findings cannot be directly transferred to the sea urchin species (S. purpuratus) used in our study, as also confirmed by Taylor and Heyland (2017, Molecular and Cellular Endocrinology). Therefore, we feel that, especially for S. purpuratus, the endogenous synthesis of THs was never fully convincing. In any case, the sentence mentioned by the reviewer refers only to our response letter, in the manuscript we never wrote such a statement. On the contrary we were extremely careful and explicitly wrote: ‘’Echinoderms represent an interesting case study. Although there is evidence suggesting the presence of an internal pathway for THs synthesis [16,22], it is most commonly believed that their source of iodotyrosines comes from diet and these are used as an indicator of nourishment [10,23].’’